# Immunoengineering via Chimeric Antigen Receptor-T Cell Therapy: Reprogramming Nanodrug Delivery

**DOI:** 10.3390/pharmaceutics15102458

**Published:** 2023-10-13

**Authors:** Theodora Katopodi, Savvas Petanidis, Doxakis Anestakis, Charalampos Charalampidis, Ioanna Chatziprodromidou, George Floros, Panagiotis Eskitzis, Paul Zarogoulidis, Charilaos Koulouris, Christina Sevva, Konstantinos Papadopoulos, Marios Dagher, Nikolaos Varsamis, Vasiliki Theodorou, Chrysi Maria Mystakidou, Nikolaos Iason Katsios, Konstantinos Farmakis, Christoforos Kosmidis

**Affiliations:** 1Department of Medicine, Laboratory of Medical Biology and Genetics, Aristotle University of Thessaloniki, 54124 Thessaloniki, Greece; katopodi@auth.gr; 2Department of Pulmonology, I.M. Sechenov First Moscow State Medical University, Moscow 119992, Russia; 3Department of Anatomy, Medical School, University of Cyprus, 1678 Nicosia, Cyprus; anestakis.doxakis@ucy.ac.cy (D.A.); ccharal@med.duth.gr (C.C.); 4Department of Public Health, Medical School, University of Patra, 26500 Rio Achaia, Greece; ioannachatzi@med.upatras.gr; 5Department of Electrical and Computer Engineering, University of Thessaly, 38334 Volos, Greece; gefloros@e-ce.uth.gr; 6Department of Obstetrics, University of Western Macedonia, 50100 Kozani, Greece; peskitzis@gmail.com; 7Third Department of Surgery, “AHEPA” University Hospital, Aristotle University of Thessaloniki, 55236 Thessaloniki, Greece; pzarog@hotmail.com (P.Z.); charilaoskoulouris@gmail.com (C.K.); christina.sevva@gmail.com (C.S.); kostaspap1995@hotmail.com (K.P.); mariosdag@gmail.com (M.D.); kosmidisc@auth.gr (C.K.); 8European Interbalkan Medical Center, 55535 Thessaloniki, Greece; nikvar83@gmail.com; 9Department of Medicine, Faculty of Health Sciences, Aristotle University of Thessaloniki, 54124 Thessaloniki, Greece; baswtheodorou@hotmail.com (V.T.); chryssa2000@gmail.com (C.M.M.); 10Faculty of Health Sciences, Medical School, University of Ioannina, 45110 Ioannina, Greece; nickkatsios@hotmail.gr; 11Pediatric Surgery Clinic, General Hospital of Thessaloniki “G. Gennimatas”, Aristotle University of Thessaloniki, 54635 Thessaloniki, Greece; kostafarmakis@yahoo.gr

**Keywords:** CAR T cell, reprogramming, immunoengineering, drug delivery

## Abstract

Following its therapeutic effect in hematological metastasis, chimeric antigen receptor (CAR) T cell therapy has gained a great deal of attention during the last years. However, the effectiveness of this treatment has been hampered by a number of challenges, including significant toxicities, difficult access to tumor locations, inadequate therapeutic persistence, and manufacturing problems. Developing novel techniques to produce effective CARs, administer them, and monitor their anti-tumor activity in CAR-T cell treatment is undoubtedly necessary. Exploiting the advantages of nanotechnology may possibly be a useful strategy to increase the efficacy of CAR-T cell treatment. This study outlines the current drawbacks of CAR-T immunotherapy and identifies promising developments and significant benefits of using nanotechnology in order to introduce CAR transgene motifs into primary T cells, promote T cell expansion, enhance T cell trafficking, promote intrinsic T cell activity and rewire the immunosuppressive cellular and vascular microenvironments. Therefore, the development of powerful CART cells can be made possible with genetic and functional alterations supported by nanotechnology. In this review, we discuss the innovative and possible uses of nanotechnology for clinical translation, including the delivery, engineering, execution, and modulation of immune functions to enhance and optimize the anti-tumor efficacy of CAR-T cell treatment.

## 1. Introduction

As one of the most dynamic areas of immuno-oncology today, CAR-T cell treatment offers patients with advanced malignancies significant advantages [1,2]. However, in contrast to hematological cancers, solid tumor treatment poses a distinct set of difficulties [3]. The lack of uniformly generated tumor-specific antigens and the immunosuppressive tumor microenvironment are the main barriers to the efficacy of CAR-T cell therapy [4]. To expand its therapeutic potential, an efficient and accurate approach for CART cells is required. A fresh alternative method for CAR-T therapy has emerged using nanotechnology. CARs are recombinant receptor structures made up of intracellular T cell signaling domains connected to a hinge/spacer and a transmembrane domain [5]. These are additionally connected to single-chain variable fragments (scFvs) of monoclonal antibodies or heavy-chain variable domains (VHHs) [6]. Major histocompatibility complex (MHC)-independent CARs directly identify antigens on cell surfaces, allowing the use of tailored antibodies for a particular antigen in patients. By targeting CD19, CD22, and BCMA antigens specifically, CAR-T treatment has received a lot of attention from biopharmaceutical companies throughout the world and has been successful in treating patients with Bcell malignancies [7]. Patients with severe Bcell acute lymphoblastic leukemia (ALL) and patients with lymphoma have experienced a very high remission rate as a result of CAR-T therapy [8]. CAR-T therapy acquired FDA approval in 2017 to treat leukemia and lymphoma, making it a clinical research achievement of cutting-edge research [9]. The viability and scalability of this therapy continue to face significant obstacles despite the high level of excitement following FDA authorization of two CAR-T therapies. First off, it is a rather complicated process to extract T cells, express CAR on them, and propagate them [10]. It calls for enormous structural resources, financial capital, and advanced clinical supervision [11]. Second, because prior treatments and aging can impair T cells’ functional status, many patients are either not eligible or suited for CAR-T therapy [12]. This may hinder the production of CAR T cells, and the injection of CAR T cells might potentially result in fatal diseases. Furthermore, despite the promising outcomes of CAR T cells against hematological cancers, the effectiveness of this treatment in solid tumors is still inadequate and confronts a number of difficulties [13]. The above-mentioned difficulties can be overcome through nanotechnology. In order to deliver immunotherapy and diagnostic drugs in a clinical environment, nanoparticles (NPs) have been firmly anchored in medical biotechnology [14]. The therapeutic chemicals are loaded into NPs in a way that effectively delivers them to desired locations of action without being hindered by immune reactions. Thus, nanotechnology can create DNA carriers that are efficient and affordable [15]. As they circulate throughout the patient, these carriers can specifically program effector cells with the ability to recognize tumors. These requirements should be met in order to construct an NP vehicle for CAR payload to modify immune cells [16]. Targeted cells must take it up, it must be stable during blood circulation, it must not trigger an immune response, it must be made to increase the efficiency of transfection in targeted cells, and it must safely navigate the cell machinery after internalization before delivering its payload to the cell nucleus [17]. To create and implement successful NP-based CAR therapy, a deeper comprehension of NP structure and the cellular mechanisms involved in NP endosomal escape as well as nuclear transport is essential [18]. In this review, we give a thorough overview of the function of nanotechnology in CAR-T treatment and emphasize its various elements. We analytically examine the obstacles to NP intracellular trafficking and discuss the design of NPs and CAR cargo in order to overcome clinical obstacles and accomplish successful nuclear delivery. The methodologies for CAR-T design and treatment are also discussed thoroughly. Finally, preclinical data from nano-based CAR-T therapies are also analyzed, in order to direct future research nanotechnology-related CAR-T approaches.

## 2. Molecular Structure of CARs

An extracellular single-chain variable fragment (scFv), a spacer/hinge region, a transmembrane region, and cytoplasmic domains compose CAR’s traditional structure [19]. CAR has progressed extensively during recent decades (Figure 1). First-generation CARs are distinguished by the existence of a single intracellular signaling domain (CD3), whereas later CAR generations offer tailored signals with the addition of one or two costimulatory endodomains to the CD3 motif to improve induction and survival [20]. Fourth-generation CARs were created by altering the expression cassette that is either constitutive or inducible and contains a transgenic protein, such as a cytokine, to further increase CAR-T cell efficiency [21].

By inserting an IL-2 receptor domain between the CD3 and CD28 signaling domains and attaching the YXXQ sequence at the C-terminal CD3, the fifth generation builds on the framework of the second-generation CAR [23]. The ability of different generations of CARs to infiltrate and aggregate in the tumor microenvironment (TME) has been made possible thanks to nanotechnology. This significantly improves the tumor-eradicating effect of CAR and reduces its negative effects [24]. The extracellular antigen-binding domain of the CAR molecule specifically recognizes the tumor antigen; the intracellular domain of the CAR molecule recognizes signal transduction; and the intracellular domain activates the T cells, causing cells to proliferate, synthesize perforin and granzyme, release cytokines, and other processes that lead to tumor cell necrosis [25,26]. The structural makeup of CAR molecules is essential for maximizing these aforementioned effects.

## 3. Current Obstacles in CAR-T Cell Therapy

CAR-T cell manufacturing is a laborious and time-consuming process that imposes a heavy financial and physical burden on patients and their families [27]. Lymphocytes taken from peripheral blood and genetically modified later become CART cells. Effective CAR-T cell expansion and maintenance in vivo are acknowledged as fundamental predictors of long-lasting clinical remissions in cancer patients [28]. T cells were consistently shown to have low persistence in clinical studies of CAR-T treatment with poor overall efficacy [29]. When CART cells are utilized to treat solid tumors, they are first circulate in the blood, with some of them penetrating tissues, draining through lymphatics, or dying in place [30]. Few can get to the tumor locations. Additionally, solid tumor stroma can prevent CAR-T cell penetration. Even if CART cells manage to reach tumor locations, they have to face the hypoxic and highly immunosuppressive TME [31]. When T cells enter the TME, immunosuppressive tumor-associated cells, such as tumor-associated macrophages (TAMs), MDSCs, cancer-associated fibroblasts (CAFs), and Tregs will contain them and suppress their functions [32]. These cells also express inhibitory molecules, such as CD80/CD86, which bind the inhibitory receptor CTLA-4 and produce soluble substances that inhibit or trigger apoptotic death in T cells. Additionally, tumor cells produce ligands such PD-L1 and Gal9 that bind to the T cell inhibitory receptors PD-1 and TIM-3, respectively [33]. The lower efficacy is caused by all of these circumstances, which encourage an “exhausted” phenotype in the CART cell. Co-administration of antibodies that block PD-1 and other inhibitory pathways has been shown by numerous studies to improve the effectiveness of CAR-T treatment [34]. Rapid and long-lasting clinical responses from CAR-T therapy have the potential to result in serious or even fatal side effects. Toxicities resulting from cytokines, related to systemic use of cytokines for CAR-T cell activation and TME modification, as well as the ensuing high levels of cytokines released by activated CART cells, are the first adverse event [35]. These toxicities may be controlled with targeted delivery of nanoparticle drug delivery systems and sustained-release effect (Figure 2) [36].

Strong contacts between tumor cells and host immune cells might induce CART cells to become activated and grow, which can result in cytokine-release syndrome (CRS) [38]. In numerous clinical trials, including the successful clinical trials of CAR T cells targeting CD19, individuals treated with CART cells have experienced severe and occasionally fatal cytokine storms [39]. The second is the occurrence of on-target, off-tumor effects brought on by the interaction of the CAR with the target antigen produced by normal cells [40]. The majority of CAR-T targets have an expression pattern with healthy tissues, leading to “on-target/off-tumor” toxicities by interaction of target antigens on healthy tissues. The first patient to receive CAR-T cell therapy for HER2 produced lethal toxicity due to insufficient HER2 expression in the ’s lung epithelial cells, which can be clearly detected using CART cells [41]. After receiving CART cells, the patient had respiratory distress syndrome, significant pulmonary infiltration, and passed away a few days later.

## 4. Nanotherapy-Related CAR Therapy

To overcome the difficulties associated with CAR-T treatment for solid tumors, a variety of nanotechnologies are being explored, including hydrogel, nanoparticleconjugation, transient CAR expression in T cells through RNA delivery, and others [42]. These nano-integration therapies can effectively eradicate both primary solid tumors and metastases while generating a strong anti-tumor immune response. By providing a platform for rapid development in gene therapy, mRNA treatments have the potential to transform modern medicine [43]. Modified nucleosides, synthetic capping, the insertion of extended poly-A tails, codon optimization, and other techniques can all be used to stabilize therapeutic mRNAs. Alternative therapies for CAR-T cell immunotherapy based on lipid nanoparticles (LNPs)-mRNA have come to light [44]. Since there is no chance of genetic integration, LNP-mRNA CARs can be generated, and their expression can be controlled in cell-free systems, making it safe for therapeutic applications [45]. The secret to the LNP system’s delivery function is ionizable cationic lipids. Because mRNA strands are negatively charged, they are bound inside the LNP by positive and negative electrical attraction. This enhances mRNA stability in vivo and protects it from lysosome degradation [46]. After being absorbed by cells, the LNP will fuse with the low-pH environment of the nuclear endosome and release mRNA into the cytoplasm. The LNP-based mRNA system has great in vivo transfection efficiency and reduced cytotoxicity in various cell types [47]. In 2020, Billingsley et al. made a ground-breaking LNP design for ex vivo mRNA delivery to human T cells. In order to design LNPs, they create dionizable lipids, which they then tested for mRNA delivery. The most effective LNP, C14-4, significantly reduced cytotoxicity and dramatically induced CAR expression and mRNA efficacy [48]. This approach was successful in eliciting strong tumor-killing activity and demonstrated the viability of LNP-based mRNA CAR-T cell creation techniques. In addition, a novel method for in vivo programming macrophages that can intrinsically infiltrate solid tumors into CAR-M1 macrophages with improved cancer-directed phagocytosis and anti-tumor activity showed similar results (Figure 3). In vivo administered nanocomplexes, comprising macrophage-targeting nanocarriers and plasmid DNA-expressing CAR-interferon, can create CAR-M1 macrophages that are capable of anti-tumor immunomodulation, CAR-mediated cancer phagocytosis, and suppression of solid tumor growth [49]. Additionally, mRNA LNPs for cytokines and vaccines were created to support or improve CAR-T treatment. For instance, researchers created IL-12 nanostimulator-engineered CART cells with biohybrids because IL-12 has potent anticancer effects and can drive the formation of tumor-specific T cells [50]. By considerably promoting the production of CCL5, CCL2, and CXCL10, the released IL-12 can further increase the recruitment and growth of CD8^+^ CART cells in the tumor. In the end, the immune-stimulating action of the IL-12 nanochaperone dramatically improved the anti-tumor activity of CART cells, eliminated solid tumors, and reduced side effects. In the meantime, the CAR-T and mRNA vaccine combination (CarVAC) was tested in a Phase I/II clinical trial (NCT04503278), and both treatments, which target the cancer-associated antigen claudin-6, were found to be safe and well-tolerated in a group of patients with solid tumors [51].

## 5. Nano-Induced T Cell Editing

Nanotechnology also makes T cell editing for CAR-T creation easy, fast, and precise. Genes in lymphocytes may often be silenced using a megaTAL nuclease-coding mRNA [52]. However, it is also possible to effectively impair T cell receptor expression on lymphocytes using the CRISPR CAS9 system [53]. Reducing the expression of the T cell receptor (TCR) can be used to produce allogenic CART cells from a healthy donor [54]. To effectively disrupt the TCR, several approaches have been developed. Recently, Moffet et al. created mRNA NPs that “hit-and-run” when combined with target cells, meaning that their mRNA cargo is quickly picked up by expressed T cells (Figure 4) [55]. CART cells’ genomes are edited using the lymphocyte-targeted mRNA NPs without altering their functionality. Nuclease-encoding mRNA NPs can change the expression of the TCR of cells without affecting CAR integration or cell proliferation. Additionally, the HSC genomes’ permanent incorporation of self-renewal genes is made appealing by the transitory expression provided by mRNA particles. Researchers created NPs that target CD105 with mRNA containing the Musashi-2 self-renewal gene [56]. When CD105 was exposed to these NPs, there was an induction in Musashi-2 expression as well as a greater number of cell surface markers pinpointing increased regenerative prospective. Preceding research indicated that the inability to grow HSCs ex vivo without triggering differentiation is a barrier to their viral integration in therapeutic application [57]. Nevertheless, a mRNA-induced NP expression can induce primitive HSC ex vivo proliferation and regeneration [58].

## 6. In Vivo Reprogramming of CART Cells

An innovative and pioneering method for streamlining and standardizing the difficult ex vivo CAR-T cell manufacturing process is in vivo programming of CART cells using nanoparticles [59]. The systemic toxicity of CRS and ICANS (immune effector cell-associated neurotoxicitysyndrome) is significantly decreased by the in situ production of CART cells. Researchers recently achieved in vivo induction of CART cells using nanodelivery of CAR components and gene-editing tools [60]. By injecting nanoparticles containing CAR-DNA or CAR mRNA, respectively, they were able to achieve the stable and transitory production of the targeted CAR protein in T cells [61]. In recent studies, the second-generation CAR structure that was specifically aimed at CD19 was combined with a cationic polymer called poly(bamino ester) to form the core of the nanodelivery system [62]. Anti-CD3 antibody-conjugated polyglutamic acid (PGA) made up the exterior of the nanodelivery carrier. In many cases of immune failure, inhibition of in vivo methylation boosts T cell rejuvenation while de novo DNA methylation causes T cell exhaustion. Decitabine, a DNA methyltransferase inhibitor licensed for therapeutic use, can alter DNA methylation patterns related to exhaustion. For this reason, decitabine-treated chimeric antigen receptor T (dCAR T) cells can exhibit improved anti-tumor activity, cytokine production, and proliferation both in vitro and in vivo. Furthermore, dCAR T cells can effectively mount recall responses when the tumor is reactivated and destroy enlarged tumors at a modest dose [63]. Higher expression levels of memory-, proliferation-, and cytokine-production-related genes are induced in dCAR T cells via tumor-producing antigen cells. In vivo, tumor-infiltrating dCAR T cells maintain a disproportionately high expression of genes associated with memory and a low expression of genes associated with exhaustion. Furthermore, recently, a pioneering approach for light-switchable CAR (LiCAR) T cells enables precise real-time phototunable activation of therapeutic T cells. LiCAR T cells enable both spatial and temporal control over T cell-mediated anti-tumor therapeutic activity in vivo with significantly reduced side effects when combined with imaging-guided, surgically removable nanoplates that have enhanced near-infrared-to-blue upconversion luminescence as deep tissue photon transducers [64]. This nano-optogenetic immunomodulation technology paves the way for the development of precision medicine and individualized anticancer therapy in addition to offering a novel method for examining CAR-mediated anti-tumor immunity [65].

## 7. Future Challenges

Regardless of the increased level of significance, there are still some difficulties and obstacles to conquer in CAR-T cell therapy. Off-target signaling is a concern associated with the in vivo administration of nano-based CARs [66]. Unintended gene transfer into HSCs carries a significant danger because gene therapy trials have previously shown that HSCs can change malignantly and cause leukemia [67]. In some circumstances, severe neurotoxicity and CRS after CAR-T therapy can be lethal, and managing these toxicities continues to be very difficult. No such incident has been reported in preclinical research including NP-based CAR treatment, but this would require careful thought and precaution when it comes to clinical studies later on [68]. Furthermore, prior research has demonstrated that the immune system may identify NP molecules as foreign substances, triggering an immune reaction through a convoluted course of action [69].

This could start CRS, which could trigger more severe on-target or off-target toxicity. Recently, a novel delivery method was developed to increase the “attack power” of CAR T cells against solid tumors. Researchers created a hydrogel with a specific construction that briefly “wraps” CART cells and stimulatory cytokines together. The gel is “applied” to the tumor using an injection needle, where T cells can develop and proliferate, creating an ongoing flow of lymphocytes to eradicate the tumor cells (Figure 5) [70]. However, it is necessary to conduct clinical studies to fully understand the toxicities associated with NPs and CARs (particularly CRS and neurodegenerative disorders) [71]. Additionally, several in vivo toxicities associated with NPs have been documented, such as off-target accumulations of particles, liver damage, oxidative stress, neutrophil activation, and oxidative stress [72]. To prevent on-target and off-target toxicities, it is essential to understand how NPs behave in vivo while interacting with the immune mechanism. It is yet unclear whether the increased immunological activation caused by NPs would also increase autoimmune toxicity and adverse events, as well as what strategies will be needed to combat these occurrences [73]. According to recent data, NP may reduce functional T cell proliferation and the proportion of CD4^+^ to CD8^+^ cells [74]. T cells constitute the foundation of CAR-T therapy; as a result, careful design tactics are essential for manipulating such cells, and a thorough characterization of the toxicity profiles of NPs is required. The reprogramming of damaged immune cells is another potential obstacle for the production of CART cells in vivo [75]. Senescence or exhaustion in T cells can be caused by prolonged contact with tumor antigens, age, or intensive pre-treatment [76,77]. These are practical difficulties with the in vivo engineering of T cells that CAR-T treatment must overcome. However, the relationship between NPs and the immune system is debatable and extremely contentious. Conversely, continuing the follow-up of patients treated with CAR T cells has little to no toxicity and has shown promising results [78,79]. In many cases, nanotechnology can address these immunotherapy issues by creating nanocarriers that carry carefully selected immunomodulatory drug combinations into the TME without causing negative systemic effects (Figure 6). For example, lipid NPs can block suppressor cells in the surroundings of solid tumors and reshape the TME. NPs were created by coating them with the tumor-targeting peptide iRGD and loading them with the T cell stimulant a-GalCer, a PI3K inhibitor, and an agonist of tumor cells that suppress the immune system. It can drastically shift the TME from a suppressive to a stimulatory state when injected into the in vivo mouse 4T1 model. This method created a therapeutic cascade that allowed CART cells to enter the tumor, grow rapidly, and successfully treat the patient [80].

## 8. Conclusions

In the last decade, CAR-T has advanced significantly, with the introduction of new therapies. Parallel to this, nanotechnology has also arisen in immuno-oncology as a functional and efficient drug delivery platform, which can expand the limitations of modern CAR-T therapy. The cost of the cell modification process, increased toxicity, and barriers to CAR-T cell therapy for solid tumors are some limitations of the CAR-T therapy. Nevertheless, nanotechnology can strengthen CAR-T in this area and optimize its clinical translation. Since recent years, NP-based CAR-T has emerged as a cutting-edge and effective clinical therapeutic strategy (Table 1) thanks to its ability to generate affordable DNA carriers that can quickly and precisely edit T cells with CAR DNA. This generates enough CART cells in vivo and causes tumor regression with efficacy that is comparable to that of the traditional CAR-T therapy created ex vivo. The translation of this method, however, depends on identifying cellular barriers to clarify the intricate trip and mechanistic processes inside NPs and CAR cargo.

Due to the complex and suppressive tumor microenvironment, CAR-T treatment had limited clinical success against solid tumors. Finding the appropriate antigen on chemoresistant tumors and creating pathways for CART cells to enter and survive in the solid TME require combining CAR-T therapy with other methods. Recently, a pioneering method for base-edited CAR7 T Cells using CRISPR technology for relapsed T cell acute lymphoblastic leukemia (ALL) was produced (Figure 7) [81]. Furthemore, nanotechnology can also be used as a productive strategy in combination with CART cells to inhibit tumor-related chemoresistance and immunosuppression. In addition to replacing or competing with the traditional viral method, NP-based CAR may also be able to circumvent some limitations in current treatment models.

As a result, current nano interventions in CAR-T cell therapy have concentrated on transfection of tumor-specific CARs in T cells, immunomodulation of TME using cytokine delivery, reduction in off-target toxicity, and enhanced infiltration and prolonged persistence either by inducing memory cell production or antigen spread. Although only a few studies have examined NP-based CAR-T therapy to date, strong preclinical research indicates that it has the potential to revolutionize the treatment of malignant disorders. The current anti-tumor strategies may be replaced by or compete with NP-based CARs, which may also be able to circumvent some of the drawbacks of existing therapy models [82,83]. The development of powerful CAR-T cell therapy and the availability of targeted cancer therapies are made possible by the genetic and functional alterations encouraged by nanotechnology [84]. Due to the fact that CART cells recognize and eradicate tumor cells independently of the MHC, some of the main ways that tumors evade MHC-restricted T cell recognition, such as the downregulation of human leukocyte antigen (HLA) class I molecules and improper antigen processing, have no effect on the target cell recognition. These engineered redirected T cells are based on the generation of modified T cells with pre-defined specificity to target almost all tumor cells. Some nano-related methods are transitioning from preclinical to clinical trials with the same vigor as cancer immunotherapy. Although many of these nanoimmunoengineering techniques have shown promise for an effective CAR-T cell treatment in solid tumors, a sizable number of trials are needed to yield sufficient clinical outcomes. Clinical translation of these nanoimmunoengineering techniques is currently under investigation. For that reason, it is necessary to conduct additional research to gain a better understanding of the interaction of nano-related CARs with the immune mechanism and comprehend the functional status of immune cells prior to CAR therapy. More investigation will make a nano-based CAR strategy more compelling and speed up clinical trials for cellular therapies based on nanotechnology. This would reduce the cost and improve the efficacy of genetically altered cell therapy while also significantly streamlining the manufacturing process of cell-based treatment in a clinical setting. Biomedical engineering is crucial to the success of cancer immunotherapy, and we think that nanotechnology will soon bring a major breakthrough for CAR-T therapy.

## Figures and Tables

**Figure 1 pharmaceutics-15-02458-f001:**
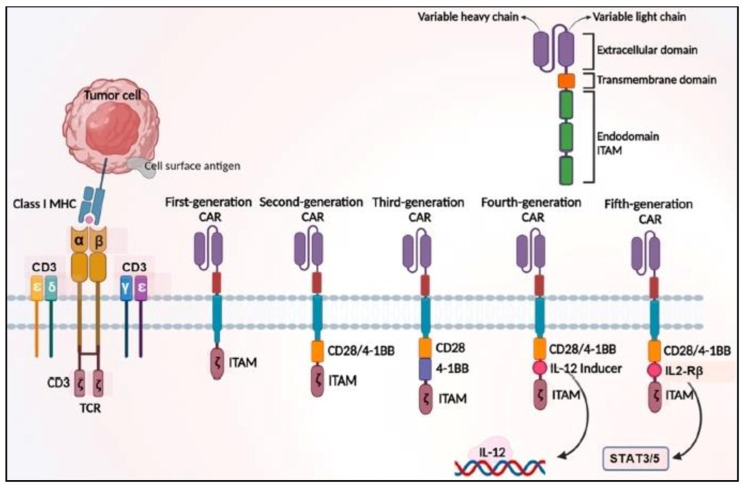
The evolution and structure of MHC-independent CARs. Monoclonal antibodies and T cell receptor complexes are the sources of CAR, forming the ectodomain of the CAR by linking the hinge domains (HD) and combining the VH and VL of the antibody (ScFv). The entire CAR is anchored to the donor or autologous T cell membrane via the transmembrane domain. The CD3-chain, which is the first-generation CAR and contains three ITAMs, is the main source of the intracellular signaling domain. In addition, each of the CD3ε-chains, carries one ITAM. The CD28 CAR of the second generation introduces a co-stimulatory signal into T cells. The T cells of the third-generation CAR (4-1BB and CD28 CAR) integrate two unique co-stimulatory signals. The fourth-generation CAR incorporates cytokine signaling into T cells as well as a co-stimulatory signal. The fifth generation of CAR has modified the intracellular binding sites for STAT3 transcription factor and another attachment region for theIL-2 receptor. Reproduced with permission from Ref. [22]. Copyright 2021, Elsevier Ltd.

**Figure 2 pharmaceutics-15-02458-f002:**
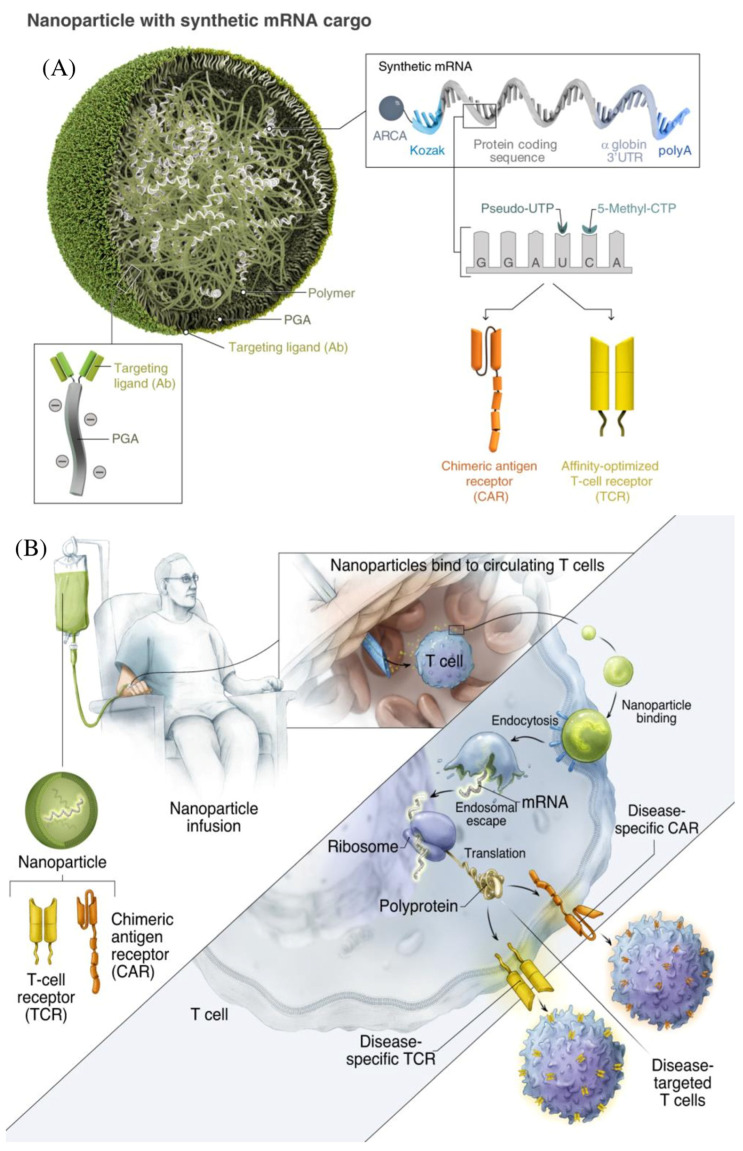
Creation and delivery of CAR mRNApolymeric nanoparticles with tumor specificity. (**A**) Bioengineered polymeric NPs are effectively able to bind to T cells via targeting ligands (Ab). (**B**) Upon their infusion, there is a brief programming for the expression of CARs that are specific to the tumor. Reproduced with permission from Ref. [37]. Copyright 2020, Springer Nature.

**Figure 3 pharmaceutics-15-02458-f003:**
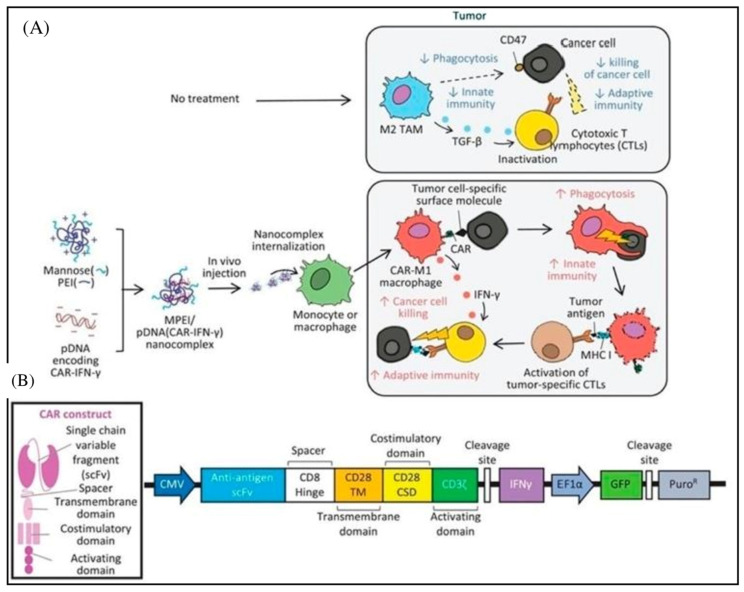
MPEI/pCAR-IFN nanocomplex therapeutic mechanisms. (**A**) Diagram showing the delivery of gene combination encoding an ALK-specific CAR and an IFN-plasmid DNA (pCAR-IFN) via MPEI to activate CAR-M1 macrophages in vivo and their anti-tumor properties. (**B**) Transgene construct including both an anti-ALK CAR and an IFN gene. Reproduced with permission from Ref. [49]. Copyright 2021, Wiley.

**Figure 4 pharmaceutics-15-02458-f004:**
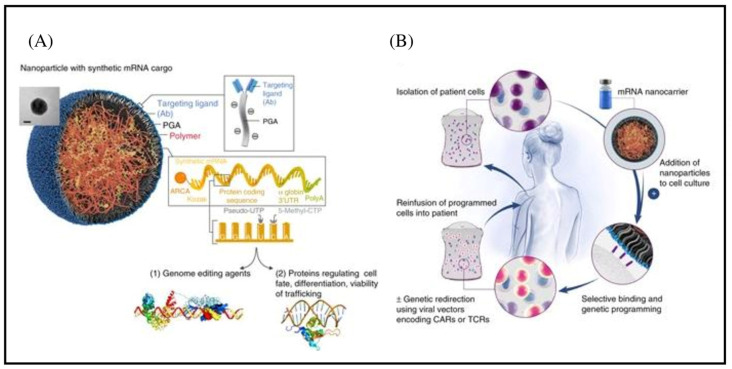
mRNA nanoparticle production for therapeutic T cell programming. (**A**) Creating specifically aimed mRNA-carrying NPs. A representative NP is depicted in the transmission electron micrograph inset. The designed synthetic mRNA contained within the NP, which encodes therapeutically important proteins, is also shown. (**B**) Schematic illustration of the programming of farmed T cells to express transgenes on polymeric nanoparticles (NPs) that are therapeutically used in patients. In order to introduce their mRNA cargoes and cause the targeted cells to express specified proteins (transcription factors or genome-editing agents), these particles are coated with ligands that direct them to particular cell types. Reproduced with permission from Ref. [56]. Copyright 2017, Springer Nature.

**Figure 5 pharmaceutics-15-02458-f005:**
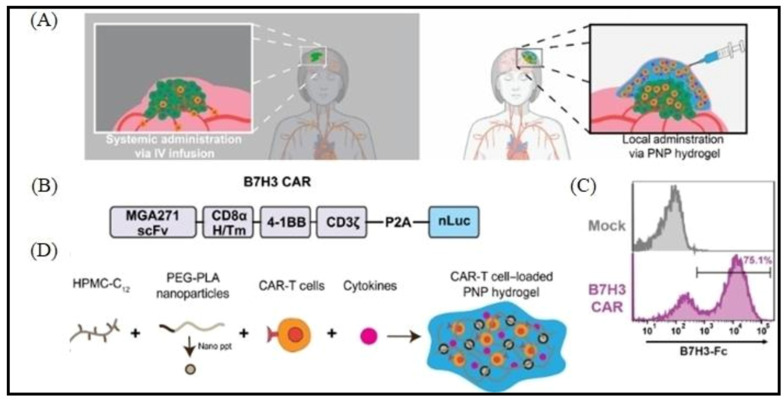
Injectable hydrogels triggering local inflammatory niche for the co-delivery of CART cells and cytokines in solid tumor therapy. (**A**) Schematic illustration comparing the intravenous (IV) administration technique for CART cells to solid tumors to existing methods. (**B**) The B7H3 CAR construct used in method is shown. (**C**) The amount of transduction that B7H3 CART cells were able to complete in comparison to untransduced “Mock” T cells after being stained with B7H3-Fc. (**D**) Dodecyl-modified hydroxypropyl methylcellulose (HPMC) and degradable block-copolymer nanoparticles self-assemble to form PNP hydrogels that co-encapsulate CART cells and stimulate cytokines. Reproduced with permission from Ref. [70]. Copyright 2022, AAAS.

**Figure 6 pharmaceutics-15-02458-f006:**
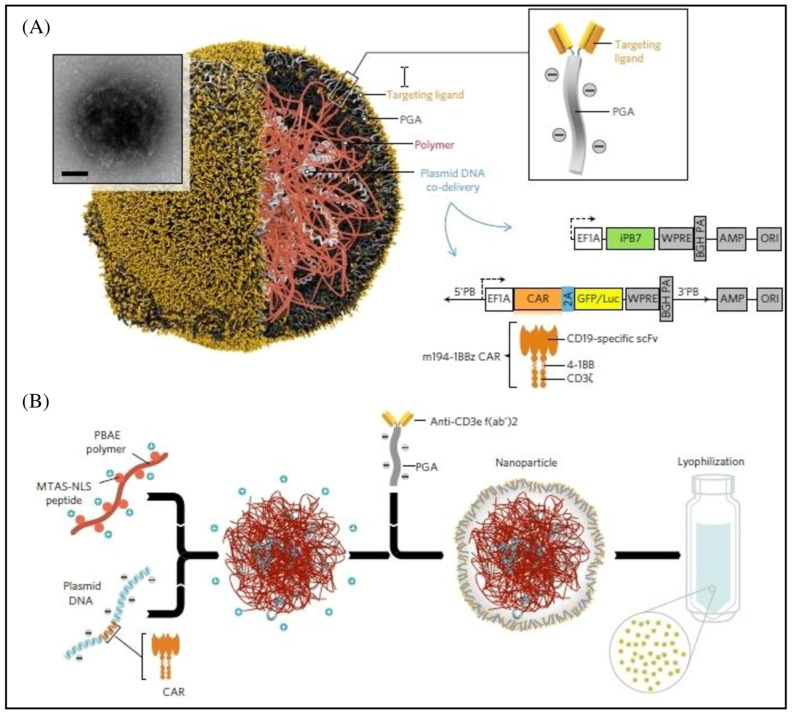
Design and production of lymphocyte-programming nanoparticles. (**A**) Schematic illustration of the T cell-targeted DNA nanocarrier. A transmission electron micrograph of a typical nanoparticle is shown. The two plasmids that were included in the nanoparticles are also shown; they contain the hyperactive iPB7 transposase and an all-murine 194-1BBz CAR. (**B**) Diagram describing the fabrication of the poly(β-amino ester) nanoparticles. Reproduced with permission from Ref. [37]. Copyright 2017, Springer Nature.

**Figure 7 pharmaceutics-15-02458-f007:**
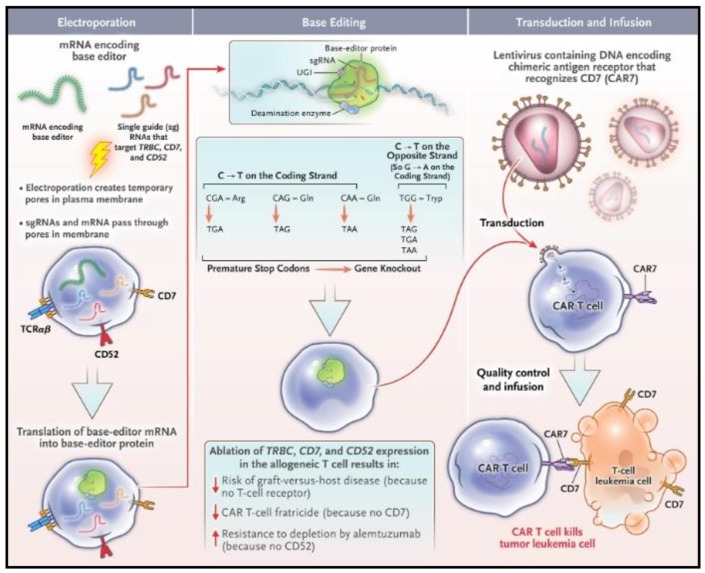
Combining CAR-T cell therapy with CRISPR method. Using donor T cells with base editing to combat T cell leukemia. Cytidine deamination gives the opportunity for highly targeted CT conversion, the addition of stop codons, or the elimination of splice sites, all of which can be used to impair gene expression without resulting in double-strand DNA breakage. By electroporating three sgRNAs against TRBC, CD7, and CD52 along with mRNA expressing codon-optimized BE3, BE-CAR7 T cells were produced from healthy donor peripheral-blood lymphocytes. This procedure made it possible to target CD7^+^ leukemia cells specifically and express CAR7 following lentiviral transduction without causing fratricide. Reproduced with permission from Ref. [81]. Copyright 2023, NEJM.

**Table 1 pharmaceutics-15-02458-t001:** Overview of NP-based CAR clinical studies for various tumor types.

Nano Type	Drug Compound	Target	Tumor Type	Status	Trial Number
Lipid NPs	mRNA-2752	OX40L, IL-23, and IL-36γ	Squamous-cell bladder carcinoma	Recruiting	NCT03739931
NPs	anti-VEGFR2	VEGFR2	Metastatic Renal Cancer	Terminated	NCT01218867
Lipid NPs	mRNA-2416	OX40L	Refractory Solid Tumor	Terminated	NCT03323398
Liposomal nanodrug	L-BLP25	MUC1	Metastatic colorectal carcinoma	Completed	NCT01462513
Liposomal nanodrug	RNA-drug	TME specific antigens	Ovarian Carcinoma	Active, not recruiting	NCT04163094
Hafnium oxide nanoparticles	NBTXR3	PD-L1	Cervical cancer	Recruiting	NCT03589339
NPs	Paclitaxel	PD-L1	Metastatic triple-negative breast cancer	Completed	NCT02425891
NPs	MEDI6469	OX40	Head and Neck Cancer	Active, not recruiting	NCT02274155

## Data Availability

Not applicable.

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
