# Peer review of "Immunoengineering via Chimeric Antigen Receptor-T Cell Therapy: Reprogramming Nanodrug Delivery"

_pharmaceutics, 2023, doi:10.3390/pharmaceutics15102458_

Round 1

Reviewer 1 Report

This review underscores the current limitations of CAR-T therapy and underscores the crucial role of nanotechnology in bolstering CAR-T's efficacy. Nanotechnology offers promising avenues for enhancing CAR engineering, augmenting T cell expansion, optimizing trafficking, boosting intrinsic activity, and modifying the immune microenvironment. These advancements hold significant promise for the development of potent CAR-T cells, ultimately improving antitumor therapy outcomes. The review carries significant importance in the realm of CAR-T cell therapy, and holds substantial promise to engage and inform its audience effectively.

Nonetheless, there are a few noteworthy areas that merit consideration in this review:

1. Incorporating recent relevant references, such as those published on the same topic (e.g., PMID: 32294554, PMID: 35401561, PMID: 32440473), would enrich the discussion and provide readers with up-to-date insights.

2.  A more comprehensive exploration of how nanotechnology can reshape the tumor microenvironment to stimulate CAR T cells would be valuable for readers seeking a deeper understanding of this aspect.

3.  Minor editorial editing is required before publication.

Line 135: time-consuming process that places a heavy financial…… Places may be replaced by imposes.

Authors may also consider making some additional minor changes: like In vivo and in vitro will be in italic

Reviewer 2 Report

The manuscript presents an interesting and timely topic of integrating nanotechnology into CAR-T cell therapy for enhanced therapeutic effectiveness. The subject matter is highly relevant given the challenges currently facing CAR-T treatments. The paper is mostly well-written. However, there are certain areas that may need further elaboration and clarification for the benefit of the reader.

1. The manuscript mentions MHC independent CARs several times but does not explain the mechanism underlying such independence. Given that this is a significant aspect of CAR-T therapies, a short paragraph elucidating how these CARs operate in an MHC-independent manner would be beneficial.

2. While the paper does a good job outlining the limitations of CAR-T therapies in treating solid tumors, there is a natural expectation for a follow-up discussion on how nanotechnology can help overcome these challenges. Corresponding examples of how nanotechnology enhances T-cell infiltration, reverses immune-suppressive environments, and mitigates cytokine storms would add considerable clarity to the manuscript.
